# Octreotide-Targeted Lcn2 siRNA PEGylated Liposomes as a Treatment for Metastatic Breast Cancer

**DOI:** 10.3390/bioengineering8040044

**Published:** 2021-04-03

**Authors:** Vrinda Gote, Dhananjay Pal

**Affiliations:** Division of Pharmaceutical Sciences, School of Pharmacy, University of Missouri-Kansas City, Kansas City, MO 64108, USA; vrindagote@mail.umkc.edu

**Keywords:** triple-negative breast cancer (TNBC), gene therapy, peptide ligand targeting, vascular endothelial growth factor (VEGF), epithelial-to-mesenchymal transition (EMT), tumor angiogenesis, tumor microenvironment, MDA-MB-231, MCF-7, MCF-12A, DSPE-PEG_2000_, cationic lipids, Vitamin E TPGS

## Abstract

Lcn2 overexpression in metastatic breast cancer (MBC) can lead to cancer progression by inducing the epithelial-to-mesenchymal transition and enhancing tumor angiogenesis. In this study, we engineered a PEGylated liposomal system encapsulating lipocalin 2 (Lcn2) small interfering RNA (Lcn2 siRNA) for selective targeting MBC cell line MCF-7 and triple-negative breast cancer cell line MDA-MB-231. The PEGylated liposomes were decorated with octreotide (OCT) peptide. OCT is an octapeptide analog of somatostatin growth hormone, having affinity for somatostatin receptors, overexpressed on breast cancer cells. Optimized OCT-targeted Lcn2 siRNA encapsulated PEGylated liposomes (OCT-Lcn2-Lipo) had a mean size of 152.00 nm, PDI, 0.13, zeta potential 4.10 mV and entrapment and loading efficiencies of 69.5% and 7.8%, respectively. In vitro uptake and intracellular distribution of OCT-Lcn2-Lipo in MCF-7 and MDA-MB-231 and MCF-12A cells demonstrated higher uptake for the OCT-targeted liposomes at 6 h by flow cytometry and confocal microscopy. OCT-Lcn2-lipo could achieve approximately 55−60% silencing of Lcn2 mRNA in MCF-7 and MDA-MB-231 cells. OCT-Lcn2-Lipo also demonstrated in vitro anti-angiogenic effects in MCF-7 and MDA-MB-231 cells by reducing VEGF-A and reducing the endothelial cells (HUVEC) migration levels. This approach may be useful in inhibiting angiogenesis in MBC.

## 1. Introduction

Breast cancer is the second leading cause of cancer death in United States (US), and it is the most frequently diagnosed cancer among US women [1]. Tumor metastasis is the major cause of more than 90% of breast cancer-related deaths [2]. The five-year survival rate for metastatic breast cancer (MBC) is only 28.21%. On the other hand, localized breast cancer has a very high (98.9%) five-year survival rate in US. Triple-Negative Breast Cancer (TNBC) is a type of breast cancer characterized by the absence of hormone receptors like estrogen and progesterone and human epidermal growth factor-2 (HER-2) receptor [2]. TNBCs can have a high rate of recurrences and systematic metastases [3]. TNBC and MBC can benefit from targeted nanotherapies, reducing tumor angiogenesis and reversing multidrug resistance (MDR) in these cells [4,5,6]. 

Current therapies of breast cancer include surgery, radiotherapy, chemotherapy, targeted therapy, and immunotherapy [7]. Small interfering RNAs are increasing explored as a new and evolving class of gene-silencing therapeutics for a variety of cancers including MBC and TNBC [8,9,10,11]. In spite of the plethora of research on siRNA therapeutics, only one siRNA product is approved in the market. Onpattro^®^ Patisiran (siRNA), marketed by Alnylam was approved by FDA in 2019 for treatment of hereditary transthyretin amyloidosis. Onpattro^®^ Patisiran is a lyophilized liposomal formulation. It is the first clinically approved RNAi therapy liposomal therapy administered intravenously [12,13]. Administering naked siRNA, cannot be useful as the ribose backbone of RNA is susceptible to hydrolysis by serum endonuclease. In addition, the small size can result in rapid glomerular filtration. These reasons are the main causes for a shot half-life less than 15 min for siRNA [14,15]. This requires delivery of siRNA through nanocarriers. Liposomes can be an ideal nanocarrier system for siRNA. They can (i) protects siRNA from degradation, thus increase circulation t1/2 (ii) entrap hydrophilic and hydrophobic cargo, (iii) low immune response, (iv) evasion of reticuloendothelial system, (v) cationic liposomes can protect siRNA from attack by endonucleases (vi) can achieve enhanced permeability and retention (EPR) effect in the tumor, (vii) can deliver siRNA to the cytoplasm, and silence the target protein to hinder tumor growth and progression, (vii) can be used for ligand-targeted delivery to the tumor cells [14,16]. An ideal siRNA nanocarrier like liposomes developed for cancer treatment, should selectively target cancer cells, and deliver siRNA to the cytoplasm. In addition to this, the therapy should silence the target protein to obstruct tumor growth and progression [17].

An important factor aiding the growth, differentiation, and eventual metastasis of tumor cells is angiogenesis. Controlling angiogenesis of tumor cells retards development of new blood vessels and shrinks the existing ones [18,19]. Anti-angiogenic agents can reduce oxygen and nutrient supply to the tumors, thus shrinking them [20,21]. Tumor cells can metastasize by the activation of epithelial-to-mesenchymal transition (EMT) pathway [22], EMT is a major contributor to metastasis of breast cancer of epithelial-originated breast cancer, like Luminal A and B type [23,24]. EMT pathway is activated during tumor cell migration, metastasis, invasion, and chemotherapy resistance [25,26]. Protein factors like matrix metalloproteinases (MMPs), epithelial markers of E-cadherin, and transforming growth factor-β1 (TGF-β1) are some of the important factors that regulate tumor cell migration and invasion in breast cancer and other types of cancer [27,28,29,30]. Hence, targeting angiogenesis and EMT pathway can be an effective solution in breast cancer metastasis. This can be useful for MBC and TNBC.

Lipocalin-2 (Lcn2) is a 25-kDa protein and a regulator of angiogenesis [31,32]. Lcn2 can induce EMT in breast cancer through estrogen receptor α/Slug axis. In addition, Lcn2 can regulate breast cancer angiogenesis. Vascular endothelial growth factor (VEGF), a key angiogenic activator, is significantly increased with higher Lcn2 expression in MCF-7 human breast cancer cells and TNBC cell line MDA-MB-436 and MDA-MB-231 cells [32]. Lcn2 levels can also be used as a non-invasive biomarker. Urine of breast cancer patients demonstrates elevated Lcn2 levels. This can correlate with progression of breast cancer [32,33,34]. Gao et al. have previously demonstrated that Lcn2 actively promote breast cancer progression via prompting EMT in breast cancer cells and by stimulating neovascularization [35]. The same group also demonstrated that Lcn2 secreted from TNBC cells increased the levels of vascular endothelial growth factor (VEGF), thus promoting neovascularization. Hence, knockdown of Lcn2 in breast cancer cells can make it an ideal anti-angiogenic target [32,36].

In the present study, we designed a PEGylated cationic liposomal formulation encapsulating Lcn2 siRNA decorated with octreotide (OCT) peptide on its surface for active targeting to breast cancer cells. OCT is an octapeptide analog of somatostatin (SST) growth hormone, having higher affinity for somatostatin receptors (SSTRs). These receptors are widely expressed on cancer cell membranes in including breast cancer [37]. SSTRs are expressed on MBC cells like MCF-7 and TNBC cells like MDA-MB-231 [38,39,40]. Here we demonstrate formulation development, optimization, and evaluation of bioactivity of OCT-targeted Lcn2 siRNA encapsulated PEGylated cationic liposomes (OCT-Lcn2-Lipo) for drug delivery to breast cancer cells like MCF-7 and MDA-MB-231.

## 2. Materials and Methods

### 2.1. Materials

Lcn2 siGENOME SMARTpool siRNA was purchased from Dharmacon (Lafayette, CO). Octreotide acetate peptide was purchased from BCN Peptides (Barcelona, Spain). Didodecyldimethylammonium bromide (DDAB), cholesterol acetate and Vitamin E TPGS was purchased from Sigma Aldrich. 1,2-distearoyl-sn-glycero-3-phosphoethanolamine-*N*-[carboxy(polyethylene glycol)-_2000_] (DSPE-PEG_2000_-COOH) and 1,2-distearoyl-sn-glycero-3-phosphoethanolamine-*N*-[carboxy(polyethylene glycol)-_2000_, NHS ester] (sodium salt) (DSPE-PEG_2000_-NHS) was purchased from Avanti Polar Lipids (Alabaster, AL, USA). *N*-hydroxysuccinimide (NHS) and 1-Ethyl-3-(3-dimethylaminopropyl) carbodiimide hydrochloride (EDC) was purchased from TCI Chemicals (Tokyo Chemical Industry, Tokyo Japan). Slide-A-Lyzer^®^ dialysis cassette (MWCO 2000 kDa) for conjugation reaction and liposome preparation and Lab-Tek^®^ II Chamber Slide for confocal microscopy was obtained from Thermo Fisher Scientific (Pittsburgh, PA, USA). Non-essential amino acids were purchased from Life Technologies^®^. Fetal bovine serum (FBS) was purchased from Atlanta Biologics (Lawrenceville, GA, USA). Quant-iT ™ RNA Assay Kit, Gibco^®^ Dulbecco’s Modified Eagle Medium (DMEM), Trypsin (TrypLE) were purchased from Invitrogen (Carlsbad, CA, USA). Dichloromethane (CH2Cl2), ethanol (EtOH), acetonitrile (AcN), and HPLC grade water (H2O) were purchased from Thermo Fisher Scientific (Pittsburgh, PA, USA). Dimethylsulphoxide D-6 for 1HNMR was obtained from Millipore Sigma. Distilled deionized (DDI) water was obtained from Barnstead EasyPure UV Deionization systems Thermo Fisher Scientific (Pittsburgh, PA, USA). Other chemicals and reagents used in this study were purchased from Sigma-Aldrich unless specified. 

#### Cell Culture

Breast cancer cell lines MCF-7, triple-negative breast cancer cell line MDA-MB-231 and normal breast epithelial cell line MCF-12A were obtained from American Type Culture Collection (ATCC, Manassas, VA, USA). Cell lines were stored in liquid nitrogen, maintained at −198 °C. T-75 Corning flask purchased from Thermo Fisher Scientific (Pittsburgh, PA, USA) were used for cell culture. Cells were cultured in DMEM (Gibco) containing 10% serum (heat-inactivated FBS), antibiotics (100 IU/mL of Streptomycin and 100 IU/mL of penicillin), 1% sodium pyruvate and 1% of non-essential amino acids. The cells were maintained at 37 °C (normal body temperature), 5% CO_2_, and 90% relative humidity in a Marshall Scientific, Heracell 150 CO2 incubator. The cells were harvested when they reached 80–90% confluence. The cell growth was observed under a ZEISS Telaval 31 Inverted Phase Contrast Microscope.

### 2.2. Synthesis of DSPE-PEG_2000_-OCT Graft Copolymer

Liposomes containing Lcn2 siRNA and surface targeted with octreotide were prepared by first targeting a particular liposomal cationic lipid with octreotide. DSPE-PEG_2000_-NHS containing a terminal-NHS group for conjugation was used for octreotide conjugation. Briefly, 3 mg of EDC, 4 mg of NHS and 10 mg of octreotide were mixed in 1 mmol of DSPE-PEG_2000_-NHS solution in PBS at pH 7.4. The reaction was carried for 12 h at dark at R.T. The conjugated lipid solution was added to Slide-A-Lyzer^®^ dialysis cassette (MWCO 20 kDa) to remove the unreacted EDC, NHS, and octreotide. The resultant polymer solution containing DSPE-PEG_2000_-NHS conjugated to Octreotide (DSPE-PEG_2000_-OCT) was lyophilized.

### 2.3. Liposomal Preparation and Optimization

Octreotide-targeted Lcn2 siRNA encapsulated liposomes (OCT-Lcn2-Lipo) were prepared by thin-film hydration method for liposome preparation. The liposomes were composed of lipids like DDAB, DSPE-PEG_2000_-OCT, DSPE-PEG_2000_-COOH, Cholesterol and Vitamin E TPGS in different molar ratios as depicted in Table 1. The liposomal formulations L-1 to L-6 were developed by changing the molar ratios of DDAB and cholesterol lipids in the polymer formulation and keeping other lipids molar ratios constant. An important independent variable of liposomal formulation like the ratio of cationic lipids to cholesterol was changed here to see its effects on the critical quality attributes (CQA’s). Molar ratios of DSPE-PEG_2000_-OCT, DSPE-PEG_2000_-COOH, and Vitamin E TPGS in the liposomal formulations were constant. This ensured all liposomes had similar OCT targeting and PEG groups on the surface for enhancing tumor targetability and circulation time, respectively. Additionally, molar ratio of Vitamin E TPGS was also kept constant. This ensured that all the formulations could have the same P-gp inhibition potential. Changing fewer parameters of the liposomal formulation enabled preparation of formulations generating more meaningful information. 

70 µmolar of the mixture of lipids was solubilized in chloroform and dried under vacuum at high speed under vacuum (Genevac, Ipswich, Suffolk, UK) for approximately 5 h. The resultant lipid film was reconstituted in 1× PBS solution using a bath sonicator. This results in the formation of multilaminar liposomes (MUV) and large laminar liposomes (LUV). A probe sonicator was used for size reduction and generation of single vesicular liposomes (SUV). At this step, 20 μg/mL Lcn2 siRNA or scrambled siRNA (SCR siRNA) was added to the liposomal mixture in PBS solution. Cationic lipids in the liposomes attract the negatively charged siRNA and thus result in encapsulation of the payload. Extrusion of nanomicellar solution was performed by 0.22 µm nylon syringe filter (Tisch Scientific, USA). Liposomal extrusion was followed by dialysis using Slide-A-Lyzer^®^ dialysis cassette (MWCO 20 kDa) for 24 h at R.T. Formulated octreotide-targeted Lcn2 siRNA encapsulated liposomes (OCT-Lcn2-Lipo) were stored at 4 °C until further use. Lcn2-Lipo without octreotide (OCT) targeting were prepared in a similar way, but instead of DSPE-PEG_2000_-OCT, DSPE-PEG_2000_-NHS was used. Placebo OCT-Lipo was made with all the above-listed polymers for OCT-Lcn2-Lipo except the addition of Lcn2 siRNA was deleted. In a similar way, Placebo Lipo without OCT targeting, and Lcn2 siRNA targeting was formulated. For imaging studies, FITC-labelled siRNA (Santa Cruz Biotechnology) was used instead of Lcn2 siRNA.

#### 2.3.1. Formulation Characterization: Size, Morphology, Zeta Potential

CQAs of OCT-Lcn2-Lipo were (i) hydrodynamic size, (ii) polydispersity index (PDI) and (iii) zeta potential and (iv) entrapment and loading efficiency. Size, PDI and zeta potential of OCT-Lcn2-Lipo was determined by Dynamic Light Scattering (DLS) Zetasizer Nano ZS (Malveran Instrument Ltd., Worcestershire UK). 700 µl of OCT-Lcn2-Lipo was placed in quartz cuvettes. Instrument was calibrated for measuring three values of hydrodynamic size and PDI. Hydrodynamic size was determined in nanometers (nm). For zeta potential, DTS1060 glass cuvettes were used. An average of three values were used for determining the final zeta potential and it was measured in millivolts (mV). The morphological characteristics of OCT-Lcn2-Lipo were visualized using Transmission Electron Microscopy (TEM). A drop of OCT-Lcn2-Lipo was placed on a copper grid sample holder. A layer of carbon and nitrocellulose was applied followed by staining the liposomes by 1% uranyl solution. The photographs were captured by a JEM 1200 EX II TEM at a voltage of 100 kV.

#### 2.3.2. Lcn2 siRNA Entrapment and Loading Efficiency

Lcn2 siRNA entrapment and loading efficiency for OCT-Lcn2-Lipo was determined by Quant-It RiboGreen RNA assay according to the manufactures’ protocol. Human Lcn2 siRNA siGENOME SMARTpool is composed of a mixture of four Lcn2 siRNAs with the following sequence: (i) D-003679-03, GAAGACAAGAGCUACAAUG, (ii) D-003679-01, GAGCUGACUUCGGAACUAA, (iii) D-003679-02, GGAGCUGACUUCGGAACUA and (iv) D-003679-05, UGGGCAACAUUAAGAGUUA. Lcn2 siRNA standard calibration curve was plotted by serially diluted Lcn2 siRNA standard solutions on a on a Spectra-MaxPlus 384 UV-Visible Spectrophotometer at an excitation wavelength of 500 nm and emission wavelength of 525 nm. OCT-Lcn2-Lipo samples were prepared by mixing the liposomes in Triton X-100 solution. This causes lysis of the liposomes and the entrapped siRNA can be measured. Briefly, OCT-Lcn2-Lipo was centrifuged at 12,000 r.p.m. for 10 min, following by discarding the supernatant. Liposomal palette was resuspended in a lysis buffer containing 1.0 mL of 0.5% Triton X-100 and vortexed in a bath sonnicator for 5 min. This solution was incubated at 37 °C water bath for 1 h. 200 µL of the lysed OCT-Lcn2-Lipo solution was mixed with 200 μL diluted Quant-It RiboGreen RNA reagent working solution (diluted 200-fold) for 10 min incubation. 200 μL of this solution was added to a black clear-bottomed 96-well plate for measuring the fluorescence. 200 μL of 1.0 mL of 0.5% Triton X-100 mixed with 200 μL 200-fold diluted RiboGreen RNA reagent working solution was used as a blank control. entrapment and loading efficiencies were calculated by the following formula:Entrapment Efficiency = (amount of Lcn2 siRNA quantified) × 100  (amount of Lcn2 siRNA added)(1)
Loading Efficiency = (amount of Lcn2 siRNA quantified) × 100  (amount of Lcn2 siRNA added) + (amount of polymers added) (2)

### 2.4. Dissolution Analysis

Dissolution study of OCT-Lcn2-Lipo and Lcn2-Lipo formulations was determined by calculating Lcn2 siRNA release from the formulations at predetermined time-points. 1000 µL of the formulations were suspended in 5.0 mL of (i) 1× PBS (pH 7.4 and pH 6.8) at and (ii) 10% FBS in 1× PBS (pH 7.4 and pH 6.8) separately, both maintained at 37 °C in a water-bath. At each time-point, the liposomal-diluted solution was centrifuged at 14,000 RPM for 20 min. 1.0 mL of the supernatant was collected, and fresh PBS was added. This helped in maintaining sink conditions, similar to physiological environment. Amount of Lcn2 siRNA released in the buffer medium was quantified by Quant-It RiboGreen RNA assay, as described for calculating OCT-Lcn2-Lipo encapsulation efficiency. 

### 2.5. Dilution Studies

Dilution of OCT-Lcn2-Lipo formulation was carried out to determine the structural integrity of the liposomal formulation on dilution. OCT-Lcn2-Lipo dilution effects were compared to Lcn2-Lipo formulation. This study was carried out in two dilution media, (i) formulation buffer at R.T. and (ii) 10% FBS in PBS at 37 °C in a water-bath. Briefly, 1.0 mL of the liposomal formulations were diluted separately with both the dilution solutions. Dilutions up to 200-fold were carried out. This was followed by analysis of liposomal size by DLS Zetasizer Nano ZS (Malveran Instrument Ltd., Worcestershire UK).

### 2.6. Stability Studies

OCT-Lcn2-Lipo was evaluated for its storage stability, stress stability and plasma stability by analyzing its size and comparing it to the non-targeted Lcn2-Lipo formulation. Storage stability involved storage of OCT-Lcn2-Lipo and Lcn2-Lipo at temperatures like 4 °C, 25 °C, and 40 °C. Samples were withdrawn at Time 0, Day 3, and Day 7. Stress stability of the two formulations was evaluated by freeze-thawing the formulations three and five times. The results were compared to time 0. Plasma stability studies involved storage of OCT-Lcn2-Lipo and Lcn2-Lipo formulations in rat plasma at the ratio of; liposomes:10% FBS 1:1 *v*/*v*. Formulations were stored at physiological temperature (37 °C) in a water bath with constant shaking. This simulated physiological conditions. Samples were withdrawn at Day 3 and Day 7, which were compared to sample taken at Time 0. These stability studies were assessed by measuring the critical quality attributes like size of the liposomes at various conditions. 

### 2.7. Cellular Uptake and Intracellular Distribution Study

Cellular uptake and intracellular distribution of OCT-Lcn2-Lipo was determined in MCF-7, MDA-MB-231 and MCF-12A cell lines. FITC conjugated siRNA was used along with Lcn2 siRNA in the liposome preparation. Similarly, non-targeted Lcn2-Lipo was prepared having FITC conjugated siRNA entrapped in it along with Lcn2 siRNA. Cellular uptake of the liposomal formulations was determined by analyzing various treated cell lines by fluorescence-assisted cell sorting (FACS). In addition to this, intracellular distribution of the liposomes was determined with confocal laser scanning microscopy (CLSM). 

#### 2.7.1. Cellular Uptake by Flow Cytometry

Cellular uptake of OCT-Lcn2-Lipo and Lcn2-Lipo containing FITC Conjugated siRNA was determined by FACS. The uptake studies were carried out in MCF-7, MDA-MB-231 and MCF-12A cell line. Briefly the cells were seeded in a 24-well plate with 5 × 10^4^ cells/well. 20 µL of OCT-Lcn2-Lipo and Lcn2-Lipo was added. All the cell lines used in this study, MCF-7, MBA-MD-231 and MCF-12A received the following three treatments; control, (serum-free media (SFM)), Lcn2-Lipo and OCT-Lcn2-Lipo. Cells were incubated for specific time points like, 1, 3, 6, and 12 h for determining uptake of various formulations. At each time point, the media was removed, and the cells were washed with Dulbecco’s Phosphate-Buffered Saline (DPBS) (Gibco’s). Cells were detached with the help of 200 µL ml of trypsin and collected in a FACS tube followed by centrifugation at 20,000 RPM for 5 min. The supernatant was discarded, and the pellet was washed twice with DPBS. This ensured removal of excess treatment groups. The final sample was made in DPBS to be used for flow cytometry. The mean fluorescence intensity of Lcn2-Lipo and OCT-Lcn2-Lipo were quantified by FACS at an excitation wavelength of 490 nm.

#### 2.7.2. Intracellular Distribution Using CLSM

Intracellular distribution of OCT-Lcn2-Lipo and Lcn2-Lipo was determined in MCF-7, MDA-MD-231 and MCF-12 cells in a time-dependent manner using CLSM. Cells were seeded at 1 × 10^4^ cells/well in an 8-chamber confocal microscopy slide (Nunc Lab-Tek II, Thermo Fisher Scientific). 10.0 µL of OCT-Lcn2-Lipo and Lcn2-Lipo was added. At each time point, the culture media was removed, followed by washing three times with DPBS (3 × 5 min). This ensured removal of excess treatment groups that were not internalized by the cells. This was followed by fixation step. Cold, buffered, 4% paraformaldehyde solution (200 µL) was added to each well. Cells were incubated at 37 °C for 20 min for fixation. After 20 min, the buffered paraformaldehyde solution was removed, and cells were washed with 200 µL DPBS (3 × 5 min). Finally, the cells were stained with mounting media containing DAPI for 15 min (Vectashield Antifade Mounting Medium). A coverslip was placed on top of the cells. The ends of the coverslip were sealed to prevent cellular dehydration and evaporation of the mounting media. Confocal microscopy slides were stored at 4 °C before imaging. Cells were observed under Leica CLSM (Leica TCS SP5, Wetzlar, Germany).

### 2.8. Cytotoxicity Study

Cellular cytotoxicity of OCT-Lcn2-Lipo, Lcn2-Lipo, placebo OCT-Lipo and placebo Lipo was determined in breast cancer cell line MCF-7, TNBC cell line: MBA-MD-231 and in normal breast epithelium cell line: MCF-12A. In vitro cytotoxicity was determined by 3-(4,5-dimethylthiazol-2-yl)-2,5-diphenyltetrazolium bromide (MTT) cell viability assay. MTT is a yellow tetrazole dye commonly used for determining cell viability. Cell lines were seeded in 96-well plates at a cell density of 1 × 10^4^ cells/well suspended in 200 µL of complete media. Cells were then incubated overnight in an incubator maintained at 37 °C in a 5% CO_2_ environment and 90% relative humidity. In vitro cytotoxicity of OCT-Lcn2-Lipo, Lcn2-Lipo, placebo OCT-Lipo and placebo Lipo formulations was determined for 24 and 72 h after initial treatment with the liposomes for 6 h. After 6 h, the media was replaced, and fresh media was added to the cells. After 24 and 72 h, cell viability was measured by MTT assay. MTT stock solution was prepared by adding MTT reagent A and reagent B in the ratio 100:1. 20 µl of MTT reagent was added to each well followed by incubation for 3 h. Absorbance of formazan solution was measured using a 96-well microplate reader at an excitation wavelength of 485 nm (BioRad, Hercules, CA, USA). A % Triton-X prepared in serum-free media (SFM) served as the positive control and SFM without any treatment served as the negative control. Cell viability was calculated according to the formula.
Cell Viability = (Absorbance of sample − absorbance of negative control) × 100  (Absorbance of positive control − absorbance of negative control)(3)

### 2.9. Lcn2 siRNA Knockdown Efficacy

Real time RT-PCR was used to examine Lcn2 mRNA expression in MCF-7 and MDA-MB-231 cells following OCT-Lcn2-Lipo treatment. Briefly, 5 × 10^5^ MCF-7 and MDA-MB-231 cells were seeded in 6-well plates and incubated for 24 h. Cells were treated (i) PBS (control), (ii) SCR, (iii) SCR-Lipo (iv) OCT-SCR-Lipo, (v) Lcn2, (vi) Lcn2-Lipo and (vii) OCT- Lcn2-Lipo at the final Lcn2 siRNA/SCR concentration of 100 nm. Cells were rinsed thrice with DPBS and further grown for 72 h. RNA was isolated from the breast cancer cells using the RNeasy Mini Kit (QIAGEN) according to the manufacturer’s protocol. cDNA was synthesized in RT-PCR process using the Reverse Transcriptase (BioRad), and levels of Lcn2 mRNA were quantified using a mixture of forward and reverse Lcn2 primers and SYBER green. Glyceraldehyde 3-phosphate dehydrogenase (GAPDH) was used as the positive control. 

### 2.10. Angiogenic Assay

#### 2.10.1. Generation of Conditioned Media (CM)

Breast cancer cells MCF-7 and TNBC cells MDA-MB-231 were treated with (i) PBS (control), (ii) SCR, (iii) SCR-Lipo (iv) OCT-SCR-Lipo, (v) Lcn2, (vi) Lcn2-Lipo and (vii) OCT- Lcn2-Lipo at the final Lcn2 siRNA/SCR concentration of 100 nm. The cells were seeded at a density of 3 × 10^5^ in a 6-well plate following addition of the treatment’s groups to the culture media. After treating the cells for 72 h, DMEM media containing the treatment groups was removed. Cells were washed twice with PBS and 1 mL of fresh serum-free DMEM was added. After 24 h, this conditioned media (CM) was removed and centrifuged at 14,000 RPM for 5 min. This helped in the removal of cellular organelles and cells. This collected supernatant was used for further in vitro anti-angiogenesis studies [1,2,3,4].

#### 2.10.2. Determination of VEGF-A Levels

CM produced by breast cancer cells MCF-7 and TNBC cells MDA-MB-231 as described above was utilized to measure the human VEGF-A levels by ELISA. VEGF-A levels in the CM was determined by human VEGF-A ELISA kit from R&D Systems (Minneapolis, MN, USA) [2]. 

#### 2.10.3. Endothelial Cell Migration Assay

Primary Umbilical Vein Endothelial Cells (HUVEC) (ATCC Mansas VA) were used for this assay. These cells were seeded on the upper chamber of COSTAR transwell filters. The transwell filters had a pore size of 8 µ m and were made from permeable polycarbonate membrane. [2,5]. The CM harvested from MCF-7 and TNBC cells MDA-MB-231 cells treated with (i) PBS (control), (ii) SCR, (iii) SCR-Lipo (iv) OCT-SCR-Lipo, (v) Lcn2, (vi) Lcn2-Lipo and (vii) OCT- Lcn2-Lipo at the final Lcn2 siRNA/SCR concentration of 100 nm was used in the is experiment. HUVEC cells seeded on the upper chamber of the trans well inserts were supplemented with CM and serum-free DMEM was added to the bottom chamber. The cells were incubated for 24 h. HUVEC migrated to the opposite/bottom side of the filter insert through the 8 μm pores were counted by Diff-Quik Stain Set. An average of four fields was counted for each sample [5].

### 2.11. Statistical Analysis

Significance in each study was determined by using at least three replicated for each experiment. The data is represented as mean ± standard deviation (SD). Statistical significance between different study groups was analyzed by two-way ANOVA or Student’s t-test. *P* < 0.05 indicated statistical significance in all experiments.

## 3. Results and Discussion

### 3.1. Synthesis and Characterization of DSPE-PEG_2000_-OCT Copolymer

Active targeting for breast cancer cells was achieved by creating an Octreotide (OCT)-targeted polymer, which was later incorporated into OCT-Lcn2-Lipo. OCT can preferably target breast cancer cells due to their high expression of somatostatin receptor [6,7]. The complete synthesis scheme for DSPE-PEG_2000_-OCT co-polymer is depicted in Figure 1. DSPE-PEG_2000_-OCT was synthesized by nucleophilic substitution reaction by addition of DSPE-PEG_2000_-NHS and OCT. The formation of DSPE-PEG_2000_-OCT was confirmed by ^1^H NMR. Figure 2 depicts the individual H^1^NMR spectrum of (A) DSPE-PEG_2000_-NHS, (B) OCT, (C) DSPE-PEG_2000_-OCT and (D) stacked spectrum of DSPE-PEG_2000_-OCT, OCT, and DSPE-PEG_2000_-NHS. In Figure 2A the short peak at 1.0–1.5 ppm is the peak for -CH_3_ of DSPE. The peak with high intensity at 1.7 ppm represents the peak for methyl groups of DSPE. The peaks at 2.7 and 3.0 ppm represents the peaks for DMSO solvent. The two peaks at 3.8 and 4.2 ppm represent peaks for methyl group in PEG for DSPE-PEG_2000_-NHS polymer. Figure 2B represents OCT H^1^NMR spectrum. The peaks at 2.5 ppm represent peaks for methyl group of lysine amino acid on OCT. While peaks at 2.7 and 4.0 ppm represent peaks for methyl (-CH_3_) and methine (-CH) group of tyrosine and tyrosine, respectively. The peak at 3.0 ppm represents the peak for DMSO solvent. The peaks from 4.5 to 5.0 ppm represent -CH groups in cysteine of OCT. Peaks from 7.5 to 8.0 ppm represent benzene rings of phenylalanine and tryptophan of OCT. Peaks with very low intensity at 9.0 ppm are due to amino group (-NH_2_) of lysine in OCT molecule. 

### 3.2. Liposomal Formulation Optimization and Characterization

OCT-Lcn2-Lipo pH-responsive liposomal formulation was composed for a mixture of five lipids forming the liposomal lipid bilayer. These were; DDAB: DSPE-PEG_2000_-OCT: DSPE-PEG_2000_-COOH: cholesterol: Vitamin E TPGS at different molar ratios as depicted in Table 1. The liposomes were prepared by thin-film hydration method. The liposomal formulations L-1 to L-6 were developed by changing the molar ratios of DDAB and cholesterol lipids in the polymer formulation and keeping other polymer molar ratios constant. An important independent variable of liposomal formulation is the ratio of cationic polymers to cholesterol. This was changed to see its effects on the CQA’s, like size, PDI, zeta potential, encapsulation, and loading efficiencies. DSPE-PEG_2000_-OCT, DSPE-PEG_2000_-COOH, and Vitamin E TPGS in the liposomal formulations were constant. This ensured all liposomes had similar OCT targeting and PEG groups on the surface for enhancing tumor targetability and circulation time, respectively. Additionally, molar ratio of Vitamin E TPGS was also kept constant. This ensured that all the formulations could have the same P-gp inhibition potential. Figure 3 depicts the schematic representation of OCT-Lcn2-Lipo.

The results formulation optimization and development by changing the independent variables like polymer molar ratios, on the dependent variables like the CQA’s of the liposomal formulation, are depicted in Table 2. Changing fewer parameters of the liposomal formulation enabled preparation of fewer formulations, generating more meaningful information. For the non-targeted liposomal formulation, Lcn2-Lipo, DSPE-PEG_2000_-OCT was replaced with DSPE-PEG_2000_-NHS. All liposomal formulations (L-1 to L-6) had size range of 95.3 to 256.6 nm, PDI of 0.13 to 0.39, zeta potential of 4.10–11.3, encapsulation efficiency of 39.3 to 96.5% and loading efficiency of 3.6 to 7.8. All the liposomal formulations had size less than 200 nm except L-4 and all had a positive zeta potential. Positive zeta potential can be attributed to cationic lipids. A positive surface charge can suggest effective internalization of the liposomes through the negatively charged cell membrane [8]. Liposomes sized 100–200 nm can be ideal for drug delivery into tumor tissue, since they can easily pass through vascular fenestrations, which are 250 nm or larger [9,10,11]. This can enable enhanced permeation and retention effect (EPR) of the liposomes [12]. With increase in size of the liposomes more than 400 nm, there are increasing chances of liposomal uptake by the reticuloendothelial system (RES) [13]. In the formulation optimization process, L-4 was not considered since it had a size higher than 200 nm. Formulation L-5 was also not considered further due to its least encapsulation and loading efficiencies. Between formulations L-1 to L-3, formulation L-2 had the least PDI and the highest encapsulation and loading efficiencies. Hence L-2 was decided as the optimized formulation having the ratio of the polymer: DDAB: DSPE-PEG_2000_OCT: DSPE-PEG_2000_-COOH:Cholesterol: Vitamin E TPGS: 50:10:10:20:10. L-2 also had the least zeta potential, but this was not considered significant factor for selecting L-2 formulation versus L1 and L-3. For all other studies, optimized formulation L-2 was used. An octreotide non-targeted liposomal formulation (Lcn2-Lipo) with lipid molar ratio same as L-2 was also prepared. Lcn2-Lipo had a size, PDI, and zeta potential of 133.7 nm, 0.16 and 5.6 mV, respectively (Figure 4).

Double chain cationic polymers like DDAB, DSPE-PEG_2000_-OCT, DSPE-PEG_2000_-COOH were utilized in the liposomal preparation. Cationic lipids aid in effective encapsulation and entrapment of negatively charged siRNA as compared to neutral lipids. Such lipids also help in achieving a net positive charge to the liposomes [14,15]. This helps in effective binding and internalization of the cationic liposomes in the negatively charged cell membrane [16,17]. Double aliphatic chain of these polymers assists in liposomal formation, which forms a lipid bilayer separating the inner aqueous environment from the outer aqueous environment. These polymers are amphiphilic in nature. The hydrophobic aliphatic chains form the inner portion of the lipid bilayer, while the hydrophobic groups form the outer portion [18]. Terminal 2000 KDa PEG group in the polymer: DSPE-PEG_2000_-COOH, has been proven to improve liposomal serum circulation time and biocompatibility [19,20]. They also prevent aggregation of the liposomes by providing a hydrophobic shield over the lipid bilayer and stabilize surface charge of the liposomes. Their property of shielding the liposome’s surface charge prevents the liposome uptake by reticuloendothelial system and degradation by serum proteins when administered in vivo [21,22]. Additionally, cholesterol increases the liposomal stability by conformational ordering of lipid chains. Cholesterol molecules are lipophilic and have a rigid planar steroid structure. They arrange themselves within the liposomal lipids and increase the packing density of the liposome [23]. Vitamin E TPGS added in the liposomal formulation acts as a P-gp inhibitor to overcome MDR in metastatic breast cancer cells [24,25,26]. Additionally, Vitamin E TPGS can also act as a stabilizing agent, solubilizer, and bioavailability enhancer [27,28,29]. Lcn2 siRNA was encapsulated within the cationic liposome by solvent evaporation-film rehydration method of liposomal preparation. Lcn2 siRNA has demonstrated reduction in lipocalin 2 protein, which is a proangiogenic factor in TNBC cell line MDA-MB-231 [2,30]. Selective targeting of metastatic breast cancer cells was achieved by targeting somatostatin receptors (SSTRs). SSTRs, are highly expressed on various cancer cells like hepatic carcinoma, neuroendocrine cancers, and ovarian and cervical carcinomas [31,32,33]. Octreotide peptide is an analog of somatostatin (SST) growth hormone, which targets SSTRs. This octreotide (OCT) can function as a targeting agent to selectively target breast cancer cells [7].

### 3.3. Dissolution Studies

In vitro dissolution for Lcn2 siRNA from OCT-Lcn2-Lipo and Lcn2-Lipo was determined by Quant-It RiboGreen RNA assay. Briefly, 1.0 mL of OCT-Lcn2-Lipo and Lcn2-Lipo was transferred was mixed with 5 mL buffer solutions: (i) PBS at pH 7.4, (ii) PBS at pH 6.8, (iii) PBS with 10% FBS at pH 7.4 and (iv) PBS with 10% FBS at pH 6.8 in separate 15 mL centrifuge tubes. Samples of the external fluid were collected at predetermined time points and Lcn2 siRNA released was determined. Figure 5A depicts the cumulative release of Lcn2 siRNA from OCT-Lcn2-Lipo and Lcn2-Lipo in 1× PBS at pH 7.4 and pH 6.8. In Figure 5A. The cumulative release of Lcn2 siRNA was higher in PBS at pH 6.8 as compared to pH 7.4. OCT-Lcn2-Lipo and Lcn2-Lipo were constructed of a pH-sensitive liposome-forming lipid: DDAB [34]. The lipid undergoes destabilization under acidic pH, resulting in faster release of the Lcn2 siRNA. There was no significant difference between the dissolution profiles of the octreotide-targeted and non-targeted liposomes, at pH 7.4 and pH 6.8 in PBS. Figure 5B depicts the cumulative release of Lcn2 siRNA from OCT-Lcn2-Lipo and Lcn2-Lipo in 10% FBS solution at pH 7.4 and pH 6.8. This study was carried out to evaluate the effect of serum proteins and pH on the dissolution profile of OCT-Lcn2-Lipo. This can be a useful tool for in vitro-in vivo correlation (IVIC) of the liposomes. PBS buffer with 10% FBS was made into two buffer solutions with different pH values (7.4 and 6.8). PBS buffer with 10% FBS at pH 6.8 can simulate conditions similar to the tumor site. The rate of Lcn2 siRNA release was faster at pH 6.8 in Figure 5B to as compared release at pH 7.4. 

### 3.4. Dilution Studies

Nanocarriers when administered in vivo undergo rapid dilution. It is important for the liposomal formulation to stay intact and deliver the cargo at the tumor site. The site of action is not the same as the site of administration. Hence, the liposomes have to travel a substantial distance to before reaching the tumor site. An ideal liposomal formulation should decrease premature drug release and be stable in the circulation. OCT-Lcn2-Lipo was diluted to 200 time in DDI water at R.T. and 10% FBS solution in PBS maintained at 37 °C water-bath with constant shaking. Size and PDI of the liposomal formulation were determined after each dilution. Table 3 depicts the results of the dilution study. 200-times dilution with DDI water at R.T. increased the liposomal size by 15.1 nm. PDI for 200-times diluted OCT-Lcn2-Lipo was higher due to possible swelling and aggregation of the liposomes. Increase in size for the serum diluted OCT-Lcn2-Lipo kept at 37 °C was higher. The difference in size between non-diluted and 200-times diluted formulation was it was 42.9 nm. However, difference in size for undiluted and 100-times diluted formulation at the same conditions was 26.0 nm. This can indicate that OCT-Lcn2-Lipo is stable upon dilution up to 100-times in serum maintained at 37 °C. 

### 3.5. Stability Studies

Temperature, freeze-thaw, and plasma stability of OCT-Lcn2-Lipo and Lcn2-Lipo was determined by measuring the size of the formulation. For temperature stability study, OCT-Lcn2-Lipo and Lcn2-Lipo were stored at 4 °C and 25 °C and 40 °C and samples were withdrawn at Day 3 and Day 7. In the freeze-thaw stability study, the liposomes were subjected to three freeze-thaw cycles and five freeze-thaw cycles. Size of targeted and non-targeted liposomes at various stress conditions was compared to size at Time 0. Figure 6 represents the results of temperature and freeze-thaw stability study for OCT-Lcn2-Lipo. There was no significant increase in size of OCT-Lcn2-Lipo and Lcn2-Lipo when stored at 4 °C and 25 °C at Day-3 and Day-7 as compared to Time 0 and when compared between Day-3 and Day-7. 

### 3.6. Cellular Uptake and Intracellular Distribution Study

#### 3.6.1. Cellular Uptake by Flow Cytometry

Cellular uptake of OCT-Lcn2-Lipo, Lcn2-Lipo containing FITC-conjugated siRNA along with Lcn2 siRNA was evaluated by flow cytometry. Uptake was determined in MCF-7, MDA-MB-231, and MCF-12A cells. This study also assisted in comparing the uptake of the OCT-targeted liposomes to the non-targeted ones. OCT-Lcn2-Lipo, Lcn2-Lipo was formulated along with FITC-conjugated siRNA as mentioned in the methods section. Time-dependent uptake was determined in all cell lines at 1, 3, 6, and 12 h. The results are depicted in Figure 7. In Figure 7A, mean fluorescence intensity (MFI) for MCF-7 cells started increasing after 3 h and a plateau in the liposomal uptake was observed from 6 h to 12 h. Highest uptake of OCT-Lcn2-Lipo and Lcn2-Lipo where a significant difference was seen in both the groups was at 6 h. Similar results were also observed for MDA-MB-231 cells (Figure 7B) and MCF-12A cells (Figure 7C). In all the cell lines, uptake of OCT-Lcn2-Lipo, Lcn2-Lipo was highest at 6 h, as indicated by the MFI. Also at this time point, we were able to see a difference in the MFI for OCT-targeted and non-targeted groups. In all the cell lines, by 12 h, most of the liposomal formulation was internalized by the cancer and normal cells. This study indicated that the uptake of OCT-Lcn2-Lipo was significantly higher than Lcn2-Lipo at 3 and 6 h, and there was a time-dependent increase in the uptake of both the formulations after 3 h, which plateaued from 6–12 h, in all the cell lines. 

#### 3.6.2. Intracellular Distribution Using CLSM Analysis

Intracellular distribution of OCT-Lcn2-Lipo and Lcn2-Lipo was determined by incubating MCF-7, MDA-MB-231, and MCF-12A cells (Figure 8) cells with OCT-Lcn2-Lipo and Lcn2-Lipo prepared with the addition of FITC conjugated siRNA for 3 and 6 h. The intracellular distribution of targeted nanomicelles was observed using CLSM. Corrected total cell fluorescence (CTCF) was calculated by Image J software to quantify and determine intracellular distribution of OCT-Lcn2-Lipo and Lcn2-Lipo in all cell lines. Figure 8A–C confocal microscopy images at 3 and 6 h and CTCF quantified for the images for various cell lines like MCF-7, MDA-MB-231, and MCF-12A, respectively. In all the figures, there is a trend of increase in time-dependent internalization of OCT-Lcn2-Lipo and Lcn2-Lipo from 3 h to 6 h. In addition, at both time points, uptake of OCT-Lcn2-Lipo was significantly higher than Lcn2-Lipo. This can be an indication, that similar to the uptake studies by FACS, OCT targeting can be a factor behind increased uptake and internalization of the liposomes. Interestingly, uptake of OCT-Lcn2-Lipois higher in the nucleus of the cells MCF-7 and MCF-12A at six hours as compared to the nucleus of MDA-MB-231 cells. This can be due to the difference in the shape of the three cell lines. MCF-7 and MCF-12A have a round shape with a round nucleus, while MDA-MB-231 are spindle-shaped cells, with a tiny round nucleus. Time-dependent increase in uptake and internalization was highest at 6 h and it was significantly higher for OCT-targeted liposomes as compared to non-targeted liposomes. These results were in accordance to the results obtained using flow cytometry analysis. 

### 3.7. Cytotoxicity Determination

Cytotoxicity of OCT-Lcn2-Lipo, Lcn2-Lipo, placebo OCT-Lipo (octreotide-targeted liposomes without Lcn2 siRNA) and placebo Lipo (non-targeted liposomes without Lcn2 siRNA) was performed on metastatic breast cancer cell line, MCF-7, TNBC cell line, like MDA-MB-231 and normal breast epithelium cell line, MCF-12A. Cells were treated with, placebo OCT-Lipo and placebo Lipo for 24 and 72 h, to evaluate the cytotoxicity of the polymers used in the liposomes. It was essential to evaluate the safety of the nanocarrier used to deliver the therapeutic cargo to the cancer cells. Evaluating the safety of the placebo formulations up to 72 h can give a better understanding of the liposomes affect cell viability. The cytotoxicity study for OCT-Lcn2-Lipo and Lcn2-Lipo, had a slightly different protocol. Here, the liposomes were added to the cells for 6 h in the cell culture media. This was followed by removal of the treatment group and addition of fresh media. Cell viability was observed for 24 and 76 h. Figure 9 depicts the cell viability (%) of all the three cell lines; MCF-7, MDA-MB-231, and MCF-12A treated with OCT-Lcn2-Lipo, Lcn2-Lipo, placebo OCT-Lipo, placebo Lipo for 24 and 72 h. Figure 9A demonstrates cell viability at 24 h above 90% and at 72 h above 85%. No significant difference between the cell viability of OCT-Lipo and placebo Lipo was seen. This can indicate that the OCT targeting did not affect the cell viability of MCF-7, MDA-MB-231, and MCF-12A cells. As seen in Figure 9B, the cell viability for breast cancer cell lines; MCF-7 and MDA-MB-231 at 24 and 72 h was approximately around 50%. This can indicate that Lcn2 siRNA was able to reduce the cellular levels of lipocalin-2 protein and thus inhibit cell growth and differentiation in the cancer cells [2]. In addition, there was a significant decrease in cell viability between OCT-Lcn2-Lipo and Lcn2-Lipo groups. This can indicate that there was a higher uptake of OCT-Lcn2-Lipo due to OCT targeting in the cancer cells, than the OCT non-targeted liposomes, in the short duration (6 h) the cells were treated. In the same figure, cell viability for normal breast cancer cells at 24 h, was more than 85%. While at 72 h, cell viability for both the Lcn2 siRNA formulations was more than 75%. This difference can be due to the doubling time of MCF-12A cells, which is approximately 36 h. There was no significant difference between the cell viability between the OCT-targeted and the non-targeted groups. 

### 3.8. Lcn2 siRNA Knockdown Efficacy

The knockdown efficacy of OCT- Lcn2-Lipo by real time RT-PCR. Lcn2 mRNA expression was measured after MCF-7 and MDA-MB-231 cells were treated with (i) PBS (control), (ii) SCR, (iii) SCR-Lipo (iv) OCT-SCR-Lipo, (v) Lcn2, (vi) Lcn2-Lipo, and (vii) OCT- Lcn2-Lipo. Figure 10A,B depicts the levels of Lcn2 mRNA in MCF-7 and MDA-MB-231 cells, respectively. MCF-7 and MDA-MB-231 cells treated with SCR, SCR-Lipo, and OCT-SCR-Lipo demonstrated no change in their Lcn2 mRNA expression levels as compared to control treatment. In Figure 10A,B, there was significant reduction in Lcn2 mRNA levels in MCF-7 and MDA-MB-231 cells treated with free Lcn2 siRNA, Lcn2-Lipo, and OCT-Lcn2-Lipo as compared to the control group in each cell line. In addition, there was a significant reduction in Lcn2 mRNA in MCF-7 cells treated with OCT-Lcn2-Lipo as compared to Lcn2-Lipo and free Lcn2 siRNA (Figure 10A). Similar results were also observed in MDA-MB-231 cell line (Figure 10B). These results could demonstrate that OCT-Lcn2-Lipo treatment could reduce the Lcn2 mRNA concentration in MCF-7 and MDA-MB-231 cells. 

### 3.9. Angiogenesis Assay

#### 3.9.1. Determination of VEGF-A Levels

VEGF is constantly secreted by tumor cells to support formation of new blood vessels to support tumor growth metastasis [35,36,37]. VEGF attracts endothelial cells from the tumor microenvironment for neovascularization [38,39]. MCF-7 breast cancer and MDA-MB-231 TNBC cells secrete VEGF-A, a sub-type of VEGF [40,41,42,43]. It is demonstrated that Lcn2 can stimulates neovascularization in breast cancer cells like MCF-7 and MDA-MB-231, and its silencing can reduce the protein levels of Lcn2 [2,4,44]. Conditional media (CM) obtained after treating MCF-7 and MDA-MB-231 cells with (i) PBS (control), (ii) SCR, (iii) SCR-Lipo (iv) OCT-SCR-Lipo, (v) Lcn2, (vi) Lcn2-Lipo and (vii) OCT-Lcn2-Lipo was used in the assay. 96-wll plated coated with VEGF-A antibody was utilized here. CM containing VEGF-A was added to this pre-coated VEGF-A wells. Amount of VEGF-A in CM is depicted in Figure 11.

#### 3.9.2. Endothelial Cell Migration Assay

The in vitro correlation of tumor angiogenesis was carried out using a transwell assay. This assay represents the ability of primary Umbilical Vein Endothelial Cells (HUVEC) (ATCC Mansas VA) to migrate towards the breast cancer cells like MCF-7 and TNBC cells like MDA-MB-231. Here, HUVEC cells were seeded on the upper chamber of COSTAR transwell filters. The transwell filters had a pore size of 8 µ m and were made from permeable polycarbonate membrane. [2,5]. The CM harvested from MCF-7 and MDA-MB-231 cells treated with (i) PBS (control), (ii) SCR, (iii) SCR-Lipo (iv) OCT-SCR-Lipo, (v) Lcn2, (vi) Lcn2-Lipo and (vii) OCT- Lcn2-Lipo was added to the lower chamber. HUVEC cells seeded on the upper chamber of the trans well inserts were supplemented with CM and serum-free DMEM. The cells were incubated for 24 h. HUVEC cells migrated to the opposite/bottom side of the filter insert through the 8 μm pores were counted by Diff-Quik Stain Set. An average of four fields was counted for each sample [5]. Figure 12 depicts the relative number of HUVEC cells migrated to the bottom chamber of the trans-well dual chamber when CM from MCF-7 (Figure 12A) and MDA-MB-231 (Figure 12C) cells was added to the bottom chamber. Figure 12B,C are the pictorial representations of the trans well assembly experimental setup for CM from MCF-7 and MDA-MB-231 cells, respectively.

## 4. Discussion

Targeting mRNA and reducing the protein expression in breast cancer using RNA interference is under intensive investigation [45,46,47]. Many such approaches primarily use liposomal drug delivery systems or lipid nanoparticles to ensure higher entrapment efficiency of siRNA [45,46,47]. The lower rate of survival for metastatic breast cancer is mainly due to its rapid progression into organs like lungs, bone, and blood. To combat this, a targeted therapy specifically targeting breast cancer cells and decreasing their angiogenic and metastatic behavior is the need of the hour. Here, we have recognized Lcn2 as a therapeutic to treat metastatic breast cancer. Lcn2 protein abnormal expression initiates the epithelial-to-mesenchymal transition (EMT) process, cell migration and invasion, and angiogenesis in breast cancer. Particularly, Lcn2 functions as an initiator of metastasis and carcinogenesis by involving multiple signaling pathways, including PI_3_K/AKT/NF-κB and HIF-1α/Erk [48]. Hence, silencing Lcn2 in breast cancer cells can help to reduce metastatic breast cancer and triple-negative breast cancer progression [4,44,49]. Here, effective delivery of Lcn2 siRNA to breast cancer cells MCF-7 and TNBC MDA-MB-231 was achieved by encapsulating Lcn2 siRNA in cationic liposomes and decorating them on the surface with Octreotide (OCT) peptide ligand. Here, we demonstrated that somatostatin receptor overexpression on MCF-7 and MDA-MB-231 cells can be used for selective delivery of cargo to breast cancer cells. Here we engineered, a liposomal drug delivery carrier for Lcn2 siRNA for its effective delivery to breast cancer cells. The liposome was targeted with OCT for selective uptake by breast cancer cells due to abnormally higher expression of somatostatin receptors on their surface.

The liposome consisted of double chain cationic polymers like DDAB, DSPE-PEG_2000_-OCT, DSPE-PEG_2000_-COOH were utilized in the liposomal preparation. Cationic lipids aid in effective encapsulation and entrapment of negatively charged siRNA as compared to neutral lipids. Such lipids also help in achieving a net positive charge to the liposomes [14,15]. This helps in effective binding and internalization of the cationic liposomes in the negatively charged cell membrane. [16,17]. Double aliphatic chain of these polymers assists in liposomal formation, which forms a lipid bilayer separating inner aqueous environment from the outer aqueous environment. These polymers are amphiphilic in nature. The hydrophobic aliphatic chains form the inner portion of the lipid bilayer, while the hydrophobic groups form the outer portion [18]. DDAB is pH-sensitive lipid that is incorporated into the lipid bilayer of OCT-Lcn2-Lipo formulation. The pH-sensitive nature aids in effective transfection of the cells by triggering an early endosomal escape of the liposomes into the cytoplasm [50,51]. Terminal 2000 KDa PEG group in the polymer: DSPE-PEG_2000_-COOH, has been proven to improve liposomal serum circulation time and biocompatibility [19,20]. They also prevent aggregation of the liposomes by providing a hydrophobic shield over the lipid bilayer and stabilize surface charge of the liposomes. Their property of shielding the liposome’s surface charge prevents the liposome uptake by reticuloendothelial system and degradation by serum proteins when administered in vivo [21,22]. PEG also provides “stealth” properties to the liposomes, which can significantly prolong the circulation time [19,52]. PEGylation strategy utilized in Doxil liposomal nanoformulation to increase circulation time [53]. Additionally, cholesterol increases the liposomal stability by conformational ordering of lipid chains. Cholesterol molecules are lipophilic and have a rigid planar steroid structure. They arrange themselves within the liposomal lipids and increase the packing density of the liposome [23]. Vitamin E TPGS added in the liposomal formulation acts as a P-gp inhibitor to overcome MDR in metastatic breast cancer cells [24,26]. Additionally, Vitamin E TPGS can also act as a stabilizing agent, solubilizer and bioavailability enhancer [27,28,29]. Lcn2 siRNA was encapsulated within the cationic liposome by solvent evaporation-film rehydration method of liposomal preparation. Lcn2 siRNA has demonstrated reduction in lipocalin 2 protein, which is a proangiogenic factor in TNBC cell line MDA-MB-231 [2,30]. Selective targeting of metastatic breast cancer cells was achieved by targeting somatostatin receptors (SSTRs). SSTRs, are highly expressed on various cancer cells like hepatic carcinoma, neuroendocrine cancers, and ovarian and cervical [31,32,33]. Octreotide peptide is an analog of somatostatin (SST) growth hormone, which targets SSTRs. This octreotide (OCT) can function as a targeting agent to selectively target breast cancer cells [7].

Guo et al. reported that intercellular adhesion molecule-1 (ICAM)-targeted Lcn2 siRNA- encapsulating liposome (ICAM-Lcn2-LP) can effectively knockdown Lcn2 mRNA and cause a significant reduction in VEGF production from MDA-MB-231 cells. In addition, ICAM-Lcn2-LP demonstrated selective uptake in TNBC cells; MDA-MB-231 as compared to normal breast epithelium MCF-10A [2]. Guo et al. also demonstrate that Lcn2 silencing, along with inhibiting the C-X-C chemokine receptor type 4 (CXCR4) can significantly reduce migration in TNBC. Here, they formulated liposomes modified with anti-CXCR4 antibodies on the surface, encapsulating Lcn2 siRNA for targeting MBC cells and for blocking the migration along CXCR4-CXCL12 axis in breast cancer. Here they achieved 88% Lcn2 silencing in MDA-MB-436 and 92% Lcn2 silencing in for MDA-MB-231 cells. Such results suggest that liposomes engineered to tackle multiple migratory pathways can be effective to slow progression of MBC [30]. Ju et al. developed and characterized OCT-modified liposomes containing daunorubicin and dihydroartemisinin as a treatment to prevent MBC. The in vitro results in MDA-MB-435S cells demonstrated higher cellular uptake and targeting by OCT-modified liposomes by OCTSSTRs (somatostatin receptors). In addition, the in vivo results exhibited a higher accumulation in tumor site along with a robust overall antitumor efficacy and negligible toxicity in MDA-MB-435S xenograft mice [7]. Zhang et al. developed octreotide (Oct)-targeted paclitaxel (PTX)-encapsulated PEG-b-PCL polymeric micelles (Oct-M-PTX) for targeting breast cancer and breast cancer stem cells. Oct-M-PTX produced the strongest antitumor efficacy, in vitro. The therapy also was found to be effective in suppressing breast cancer stem cells in vivo [6].

Here, OCT-Lcn2-Lipo was formulated and optimized to achieve maximum entrapment efficiency and lowest size. Solvent evaporation-film rehydration method was used for the formulation preparation. A total of 100 nm of Lcn2 siRNA was utilized for formulation preparation. Optimized formulation, L-2 had the least PDI and the highest encapsulation and loading efficiencies for Lcn2 siRNA. L-The ratio of various lipids in the liposome were: DDAB: DSPE-PEG_2000_OCT: DSPE-PEG_2000_-COOH: Cholesterol: Vitamin E TPGS: 50:10:10:20:10. OCT-lcn2-Lipo, the optimized formulation had a size of 152 had a size of 152.00 nm, PDI 0.13, zeta potential 4.10 mV, entrapment and loading efficiencies of 69.5% and 7.8 %, respectively. OCT-Lcn2-Lipo dilution study indicated that there was no significant difference in liposomal size when the liposomes were diluted up to 100 times in PBS solution. However, swelling and increase in size was observed when OCT-Lcn2-Lipo was incubated in 10% FBS solution diluted up to 200 times and stored at body temperature. Temperature stability indicated that OCT-Lcn2-Lipo was stable up to Day-7 at 4 °C and 25 °C. The OCT-Lcn2-Lipo demonstrated an increase in size as compared to time 0 when stored at 40 °C and when subjected to freeze-thawing. 

In vitro uptake of OCT-Lcn2-Lipo in MCF-7 and MDA-MB-231 cells using flow cytometry determined the effective internalization of the liposomes in a time-dependent manner. There was a significant difference in the uptake of OCT-Lcn2-Lipo as compared to Lcn2-Lipo at 6 h in both the cell lines. Uptake plateaued around 12 h. In vitro uptake was also determined in MCF-12 normal breast epithelium cells. The total uptake for the normal cells was lower than the uptake for the breast cancer cells. Similar results were also obtained for intracellular distribution study using confocal microscopy. In vitro cytotoxicity study depicted that the placebo liposomes were safe on all the above three cell lines. While the OCT-Lcn2-Lipo was reduced cell viability to a significantly greater extent in MCF-7 and MDA-MB-231 cells as compared to Lcn2-Lipo. This can imply that OCT targeting can result in higher accumulation of Lcn2 siRNA in the breast cancer cells as compare to the non-targeted liposomes. The cell viability was 80–90% for MCF-12A cells treated with OCT-Lcn2-Lipo and Lcn2-Lipo. This can imply that OCT-Lcn2-Lipo was safe and non-toxic on breast cancer cells.

OCT-Lcn2-lipo could achieve approximately 55–60% silencing of Lcn2 mRNA in MCF-7 and MDAMB-231 cells, respectively. Several research groups have reported the use of lipid drug delivery system for siRNA as an anti-angiogenic therapy [54,55,56,57]. To assess the anti-angiogenic potential of OCT-Lcn2-Lipo, VEGF-A levels and HUVEC cell migration assay was performed. First, all the cells were treated with OCT-Lcn2-Lipo and other treatment groups, and the media from the cells was separated. This was the CM, which was used for VEGF-A level detection and HUVEC cell migration assay. VEGF-A expression was significantly reduced in OCT-Lcn2-Lipo treatment group as compared to control group where MCF-7 and MDA-MB-231 cells were untreated. Similarly, the total number of HUVEC cells migrating toward the CM from breast cancer cells treated with OCT-Lcn2-Lipo was significantly lower than the control group of breast cancer cells. These results established the in vitro anti-angiogenic potential of OCT-Lcn2-Lipo in MCF-7 and MDA-MB-231 cells. 

## 5. Conclusions

This study represents a proof of concept for a novel MBC- and TNBC-targeted, Lcn2-silencing nanotherapy. Liposomes are a versatile platform to deliver hydrophilic cargo like gene therapy, hydrophobic small molecules, and imaging agents for breast cancer diagnosis and treatment [46]. Additionally, stimuli-responsive or “smart” liposomes are currently developed for improving the intracellular delivery of nucleic acids [58,59]. Here we engineered an OCT-targeted PEGylated liposomal drug delivery system encapsulating Lcn2 siRNA for selective targeting and for reduction of angiogenesis in MBC and TNBC cell lines.

## Figures and Tables

**Figure 1 bioengineering-08-00044-f001:**
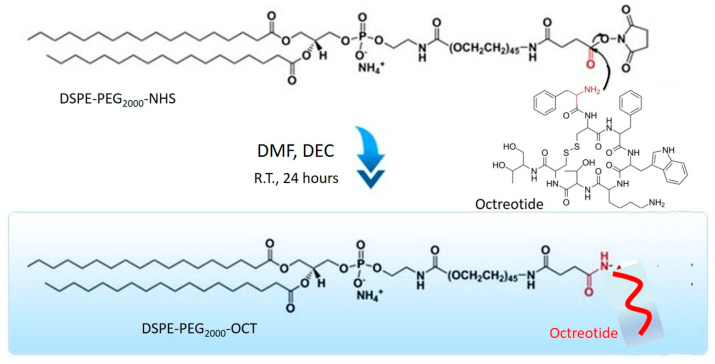
Synthesis scheme for 1,2-distearoyl-sn-glycero-3-phosphoethanolamine-*N*-[carboxy(polyethylene glycol)-_2000_] (DSPE-PEG_2000_-OCT).

**Figure 2 bioengineering-08-00044-f002:**
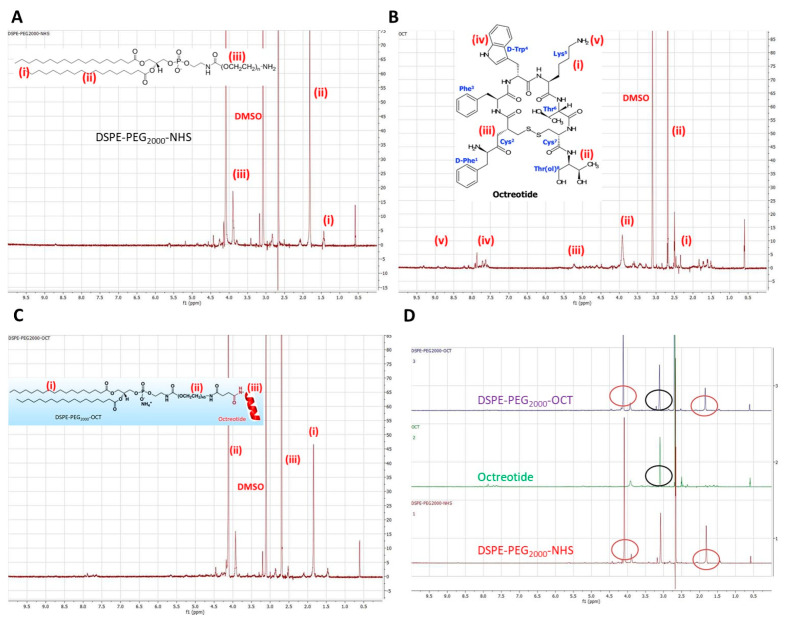
^1^H NMR spectrum of (**A**) DSPE-PEG_2000_-NHS, (**B**) OCT, (**C**) DSPE-PEG_2000_-OCT and (**D**) stacked spectrum of DSPE-PEG_2000_-OCT, OCT and DSPE-PEG_2000_-NHS.

**Figure 3 bioengineering-08-00044-f003:**
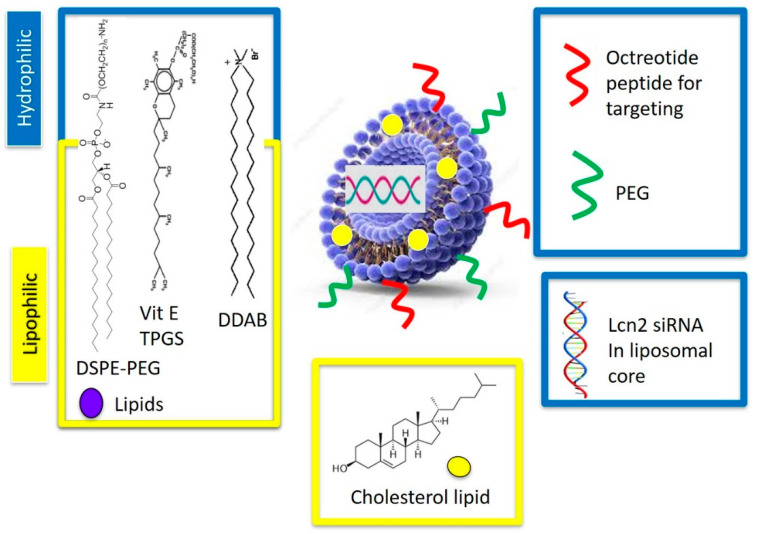
Octreotide-targeted Lcn2-siRNA encapsulated PEGylated liposomes (OCT-Lcn2-Lipo) formulation schematic representation.

**Figure 4 bioengineering-08-00044-f004:**
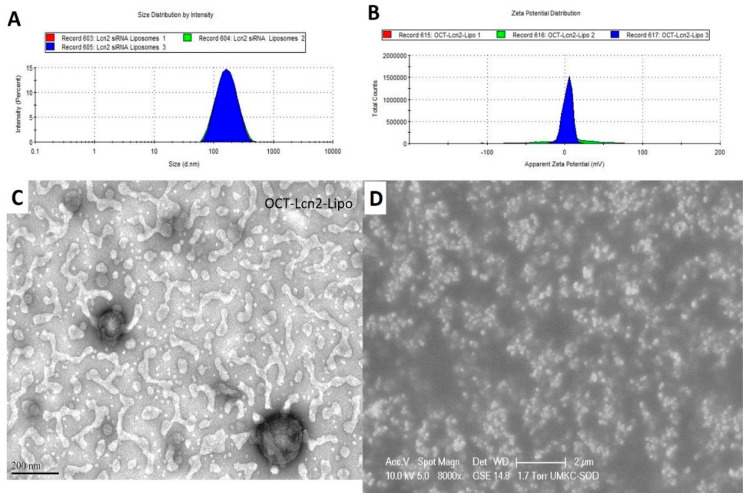
Octreotide-targeted Lcn2-siRNA encapsulated liposomes (OCT-Lcn2-Lipo) morphological characterization. (**A**) Hydrodynamic size, (**B**) Zeta potential, (**C**) Transmission electron microscopy (TEM) image of liposomal formulation in liquid state and (**D**) Scanning electron microscopy image (SEM) of liposomal lyophilized formulation.

**Figure 5 bioengineering-08-00044-f005:**
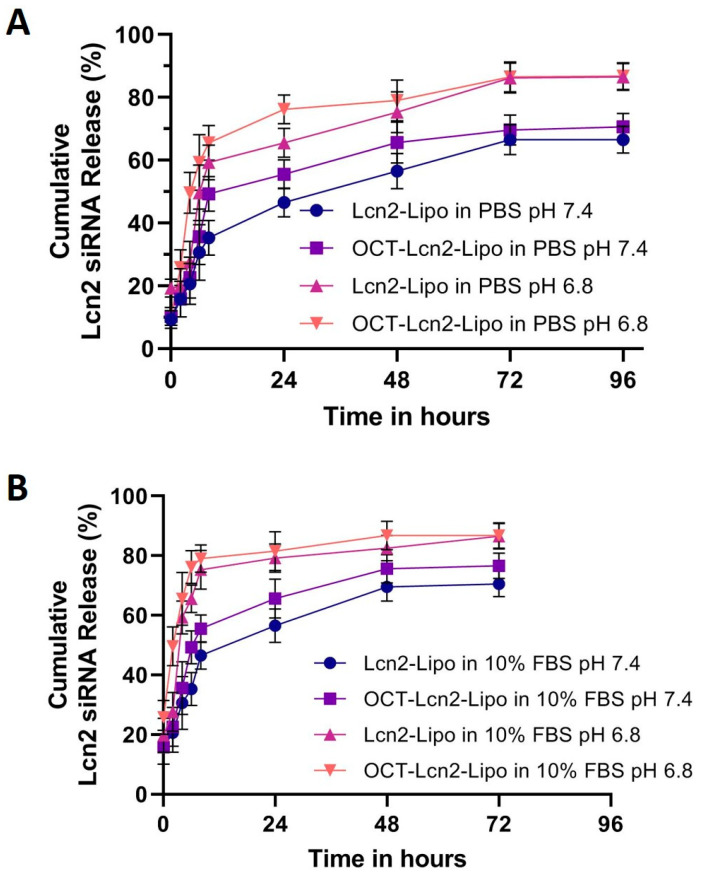
Dissolution of Lcn2 siRNA OCT-Lcn2-Lipo and Lcn2-Lipo liposomal formulations. (**A**) Cumulative release of Lcn2 siRNA from OCT-Lcn2-Lipo and Lcn2-Lipo in 1× PBS at pH 7.4 and pH 6.8. (**B**) Cumulative release of Lcn2 siRNA from OCT-Lcn2-Lipo and Lcn2-Lipo in 10% Fetal Bovine serum in 1× PBS at pH 7.4 and pH 6.8. The data were expressed as mean ± SD (*n* = 3).

**Figure 6 bioengineering-08-00044-f006:**
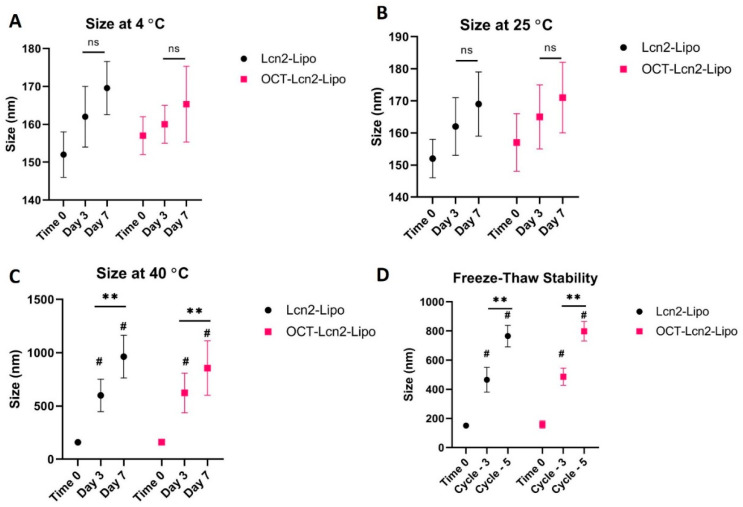
Temperature and freeze-thaw stability study for OCT-Lcn2-Lipo. (**A**) Temperature stability study for OCT-Lcn2-Lipo and Lcn2-Lipo at 4 °C, (**B**) Temperature stability study for OCT-Lcn2-Lipo and Lcn2-Lipo at 25 °C, (**C**) Temperature stability study for OCT-Lcn2-Lipo and Lcn2-Lipo at 40 °C, (**D**) Freeze-thaw stability study for OCT-Lcn2-Lipo and Lcn2-Lipo. The data were expressed as mean ± SD (*n* = 3). (# *p* ≤ 0.05 as compared to Time 0, ns = non-significant, ** *p* ≤ 0.01 as compared to Lcn2-Lipo group).

**Figure 7 bioengineering-08-00044-f007:**
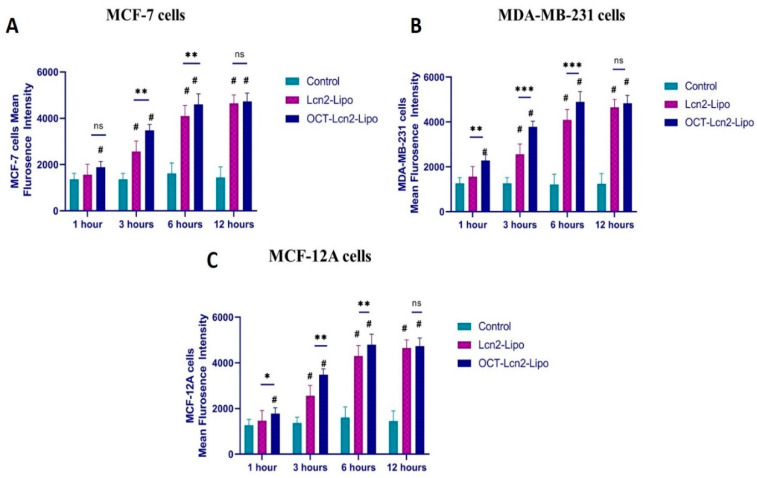
Time-dependent uptake of OCT-Lcn2-Lipo and Lcn2-Lipo prepared with the addition of FITC conjugated siRNA. (**A**) Uptake of OCT-Lcn2-Lipo and Lcn2-Lipo solution in MCF-7 cells, (**B**) Uptake of OCT-Lcn2-Lipo and Lcn2-Lipo solution in MDA-MB-231 cells. (**C**) Uptake of OCT-Lcn2-Lipo and Lcn2-Lipo solution in MCF-12A cells. The data were expressed as mean ± SD (*n* = 3). (# *p* ≤ 0.05 as compared to control group, ** *p* ≤ 0.01 and *** *p* ≤ 0.001 as compared to Lcn2-Lipo group).

**Figure 8 bioengineering-08-00044-f008:**
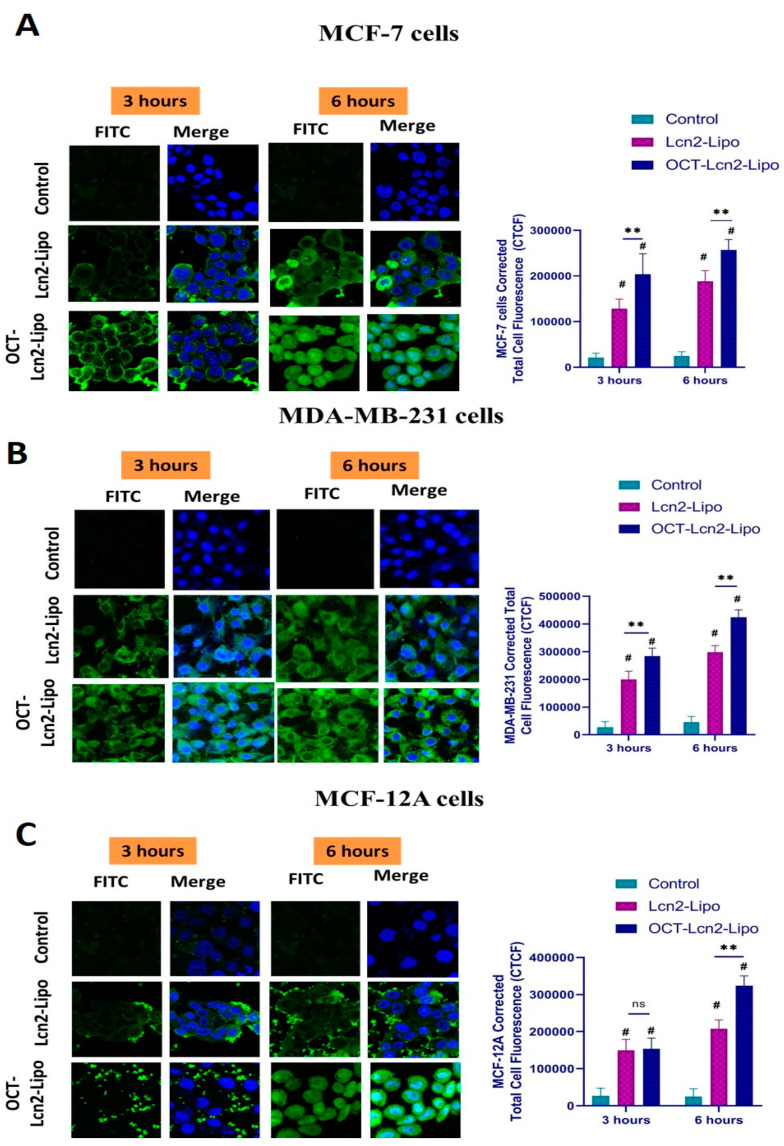
Time-dependent uptake of OCT-Lcn2-Lipo and Lcn2-Lipo prepared with addition of FITC conjugated siRNA. (**A**) Intracellular distribution and corrected total cell fluorescence (CTCF) of OCT-Lcn2-Lipo and Lcn2-Lipo formulation in MCF-7 cells, (**B**) Intracellular distribution and CTCF OCT-Lcn2-Lipo and Lcn2-Lipo solution in MDA-MB-231 cells. (**C**) Intracellular distribution and CTCF OCT-Lcn2-Lipo and Lcn2-Lipo solution in cells.MCF-12A cells. The data were expressed as mean ± SD (*n* = 3). (# *p* ≤ 0.05 as compared to control group, ** *p* ≤ 0.01 and *** *p* ≤ 0.001 as compared to Lcn2-Lipo group).

**Figure 9 bioengineering-08-00044-f009:**
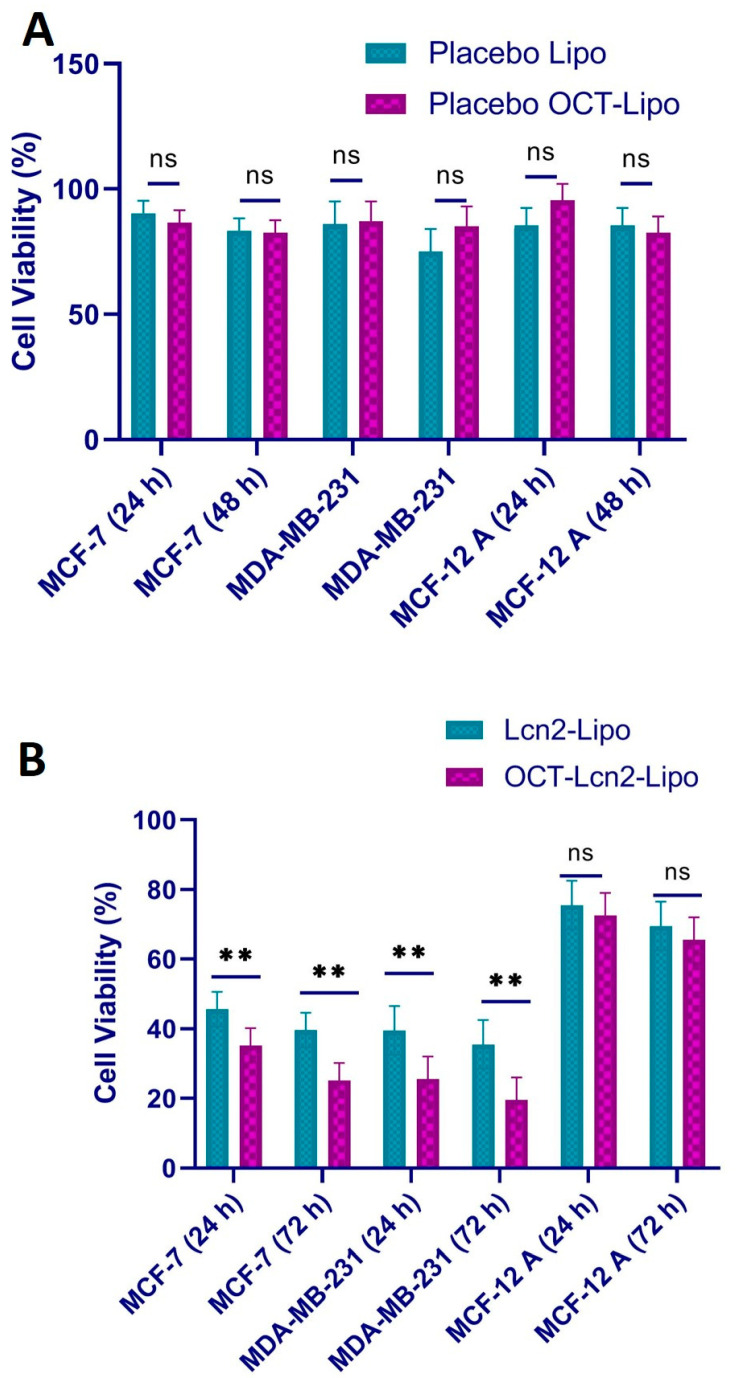
Cell viability for placebo (**A**) OCT-Lipo and (**B**) OCT-Lcn2-Lipo in breast cancer cell lines MCF-7, MDA-MB-231, and normal breast epithelium cells MCF-12A for 24 and 72 h. The data were expressed as mean ± SD (*n* = 3) (** *p* ≤ 0.01 and ns = not significant).

**Figure 10 bioengineering-08-00044-f010:**
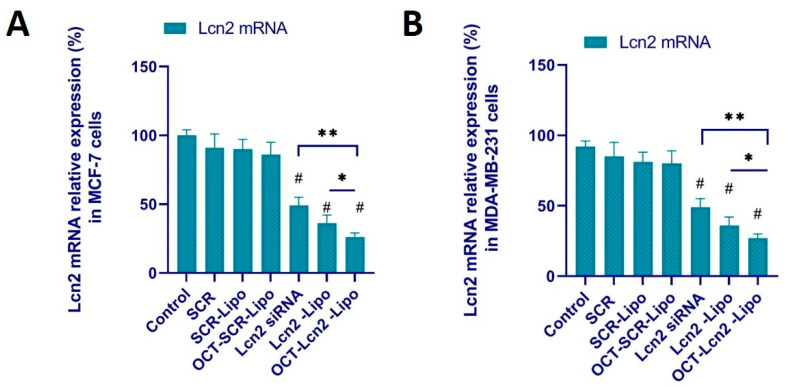
Lcn2 siRNA knockdown of Lcn2 gene expression at the mRNA level in MCF-7 and MDA-MB-231 cells using real time RT-PCR. (**A**) Lcn2 mRNA relative expression in MCF-7 cells after treatment with OCT-Lcn2-Lipo. (**B**) Lcn2 mRNA relative expression in MDA-MB-231 cells after treatment with OCT-Lcn2-Lipo. The data were expressed as mean ± SD (*n* = 3). (# *p* ≤ 0.05 as compared to control group, * *p* ≤ 0.05, ** *p* ≤ 0.01 and as compared to Lcn2 siRNA group).

**Figure 11 bioengineering-08-00044-f011:**
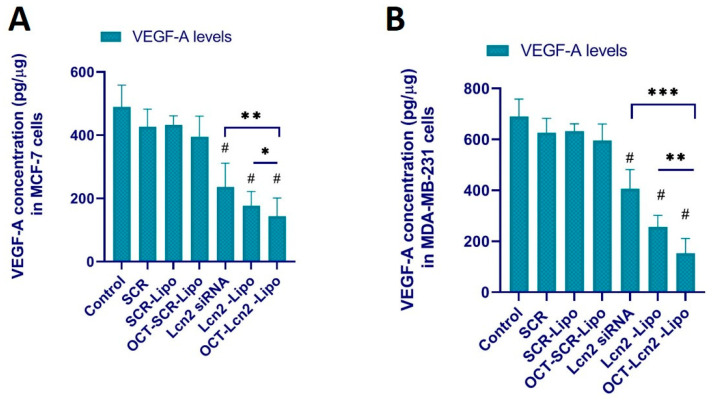
Vascular endothelial growth factor -A (VEGF-A) quantification in conditioned media (CM) from MCF-7 and MDA-MB-231 cells treated with OCT-Lcn2-Lipo using ELISA. (**A**) VEGF-A concentration (VEGF/total protein, pg/μg) in CM from MCF-7 cells treated with OCT-Lcn2-Lipo using ELISA. (**B**) VEGF-A concentration (VEGF/total protein, pg/μg) in CM from MDA-MB-231 cells treated with OCT-Lcn2-Lipo using ELISA. The data were expressed as mean ± SD (*n* = 3). (# *p* ≤ 0.05 as compared to control group, * *p* ≤ 0.05, ** *p* ≤ 0.01 and *** *p* ≤ 0.001 as compared to Lcn2 siRNA group).

**Figure 12 bioengineering-08-00044-f012:**
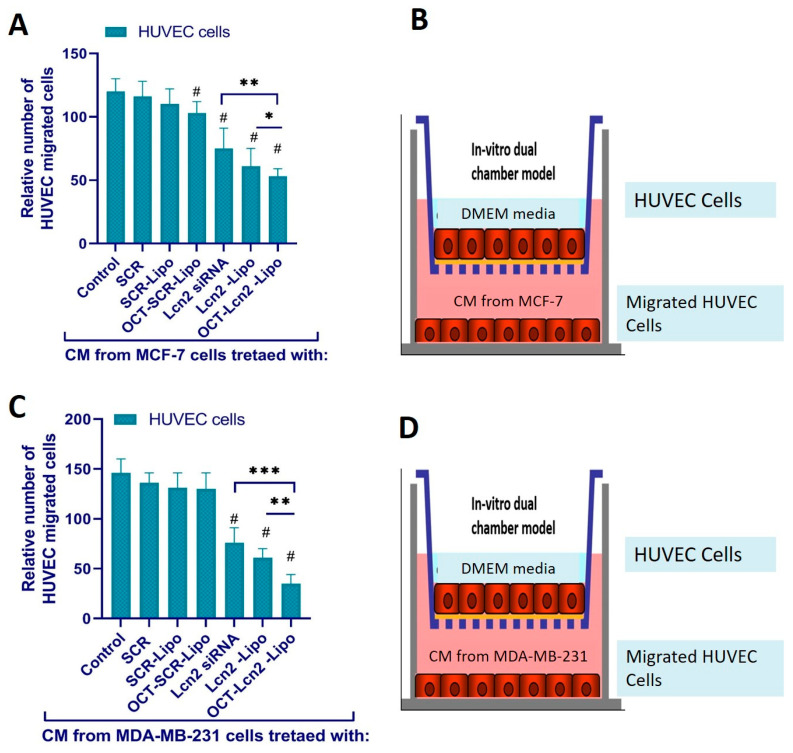
Human Umbilical Vein Endothelial Cells (HUVEC) migration toward conditioned media (CM) harvested from MCF-7 and MDA-MB-231 cells treated with OCT-Lcn2-Lipo. (**A**) Relative number of HUVEC cells migrated towards CM Figure 7. cells in a trans-well filter assembly. (**B**) Pictorial representation of Trans-well dual chamber filter assembly where HUVEC cells are in the upper chamber and CM from MCF-7 cells is in the bottom chamber. (**C**) Relative number of HUVEC cells migrated towards CM from MDA-MB-231 cells in a trans-well filter assembly. (**D**) Pictorial representation of Trans-well dual chamber filter assembly where HUVEC cells are in the upper chamber and CM from MDA-MB-231 cells is in the bottom chamber. The data were expressed as mean ± SD (*n* = 3). (# *p* ≤ 0.05 as compared to control group, * *p* ≤ 0.05, ** *p* ≤ 0.01 and *** *p* ≤ 0.001 as compared to Lcn2 siRNA group).

**Table 1 bioengineering-08-00044-t001:** OCT-Lcn2-Lipo formulation development.

Formulation Code	DDAB (Mole Ratio)	DSPE-PEG_2000_-OCT (Mole Ratio)	DSPE-PEG_2000_-COOH (Mole Ratio)	Cholesterol (Mole Ratio)	Vitamin E TPGS (Mole Ratio)
L-1	35	10	10	35	10
L-2	50	10	10	20	10
L-3	20	10	10	50	10
L-4	60	10	10	10	10
L-5	10	10	10	60	10
L-6	40	10	10	30	10
L-7	30	10	10	40	10

**Table 2 bioengineering-08-00044-t002:** OCT-Lcn2-Lipo formulation development.

OCT-Lcn2-Lipo Code. Molar Ratio of DDAB: DSPE-PEG_2000_OCT: DSPE-PEG_2000_-COOH: Cholesterol: Vitamin E TPGS	Size (nm)	PDI	Zeta Potential (mV)	Encapsulation Efficiency (%)	Loading Efficiency (%)
L-1 35:10:10: 35:10	146.3	0.21	7.5	56.4	6.1
L-2 50:10:10: 20:10	152.0	0.13	4.10	69.5	7.8
L-3 20:10:10: 50:10	112.6	0.19	10.5	45.6	3.9
L-4 60:10:10: 10:10	252.6	0.39	7.2	56.5	5.1
L-5 10:10:10: 60:10	95.3	0.26	11.3	39.3	3.6

**Table 3 bioengineering-08-00044-t003:** OCT-Lcn2-Lipo dilution study.

Dilution Factor	DDI Water Dilution @ R.T.		10% FBS Solution @ 37 °C	
	Size	PDI	Size	PDI
0	156.3	0.18	153.6	0.16
10	159.6	0.21	159.6	0.29
50	162.3	0.23	171.3	0.39
100	165.7	0.32	179.6	0.45
200	171.4	0.67	196.5	0.5

## Data Availability

Data is contained within the article.

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
