# Peer review of "Octreotide-Targeted Lcn2 siRNA PEGylated Liposomes as a Treatment for Metastatic Breast Cancer"

_bioengineering, 2021, doi:10.3390/bioengineering8040044_

Round 1

Reviewer 1 Report

Gote and Pal report synthesis and utilization of lipid-PEG-octreotide for liposomal drug delivery. As they anticipated, pegylated moiety was helpful for several aspects including stability and targeting.

This reviewer found minor points to be improved within current manuscript. 

  1. lack of information regarding octreotide because this journal is not list in pharmaceutics
  2. the peptide representation (red) in Fig1 and Fig3 is misleading. the peptide is not a helical one in structure.

I suggest the authors to perform animal experiment with the formulation in near future.

Author Response

Gote and Pal report synthesis and utilization of lipid-PEG-octreotide for liposomal drug delivery. As they anticipated, pegylated moiety was helpful for several aspects including stability and targeting.

The authors would like to thank the reviewer for their insightful and careful review of the research article.

This reviewer found minor points to be improved within the current manuscript. 

  1. lack of information regarding octreotide because this journal is not list in pharmaceutics

An additional study illustrating the advantage of Octreotide targeting is now explained in the discussions section.

  1. the peptide representation (red) in Fig1 and Fig3 is misleading. the peptide is not a helical one in structure.

This is a very accurate observation. The authors have corrected this in Figure 1 and 3 as well as in the graphical abstract.

I suggest the authors to perform animal experiment with the formulation in near future.

This is an excellent suggestion. We will be planning to perform animal studies with the current formulation or similar formulations in the near future.

Reviewer 2 Report

Authors have developed PEGylated liposomal Lcn2 siRNA and OCT-targeted Lcn2 siRNA encapsulated 14 PEGylated liposomes (OCT-Lcn2-Lipo) for targeting of MCF-7 and MDA-MB-231. They have characterized their formulation and used for uptake, proliferation and angiogenesis. This manuscript is well written and need to answer following queries

  1. Authors should write the full name of lnc2 in the abstract.
  2. Authors should remove the word in “epithelial to mesenchymal transition in and enhancing tumor angiogenesis”.
  3. “novel PEGylated liposomal system encapsulating Lcn2 small interfering”. Do you think novel is a right word in this context?
  4. In figure 8, the uptake of OCT-Lcn2-Lipo is more in the nucleus in MCF-7 and MCF-12 A but not in MDA-MB-231. Author should explain this.
  5. In figure 9 author should explain how they calculate the percentage of cell viability in figure 9 A and B. What is the control to calculate the percentage?
  6. I would suggest authors to perform tube formation assay for the angiogenesis to define the conclusion or else they have to remove angiogenesis.
  7. Why author have used condition medium from the MCF-7 and MDMBA231. Why they did not treat HUVEC cells with OCT-Lcn2-Lipo and Lcn2-Lipo for migration assay.
  8. Did the author look at the VEGF-C levels in MCF-7 as MCF-7 has high levels of VEGF-C.
  9. Authors have to improve on introduction and discussion and may be these articles will help 2020 Jun 21;9(6):1511; Cancer Epidemiol Biomarkers Prev. 2009 Feb;18(2):630-9. 

Author Response

Authors have developed PEGylated liposomal Lcn2 siRNA and OCT-targeted Lcn2 siRNA encapsulated 14 PEGylated liposomes (OCT-Lcn2-Lipo) for targeting of MCF-7 and MDA-MB-231. They have characterized their formulation and used for uptake, proliferation and angiogenesis. This manuscript is well written and need to answer following queries

The authors would like to thank the reviewer for their insightful and careful review of the research article.

  1. Authors should write the full name of lnc2 in the abstract.

This is now corrected in the revised manuscript.

  1. Authors should remove the word in “epithelial to mesenchymal transition in and enhancing tumor angiogenesis”.

This is now corrected in the revised manuscript.

  1. “novel PEGylated liposomal system encapsulating Lcn2 small interfering”. Do you think novel is a right word in this context?

This is now corrected in the revised manuscript. The word novel is removed.

  1. In figure 8, the uptake of OCT-Lcn2-Lipo is more in the nucleus in MCF-7 and MCF-12 A but not in MDA-MB-231. Author should explain this.

“Interestingly, uptake of OCT-Lcn2-Lipois higher in the nucleus of the cells MCF-7 and MCF-12A at six hours as compared to the nucleus of MDA-MB-231 cells. This can be due to the difference in the shape of the three cell lines. MCF-7 and MCF-12A have a round shape with a round nucleus, while MDA-MB-231 are spindle shaped cells, with a tiny round nucleus.”- This is explained in the text, section 3.6.2. The authors predict that this can be the reason for difference in liposome distribution internally.

  1. In figure 9 author should explain how they calculate the percentage of cell viability in figure 9 A and B. What is the control to calculate the percentage?

This is explained by the authors in section 2.8. 1% Triton-X prepared in serum free media (SFM) served as the positive control and SFM without any treatment served as the negative control

  1. I would suggest authors to perform tube formation assay for the angiogenesis to define the conclusion or else they have to remove angiogenesis.

This is an excellent suggestion by the reviewers. Unfortunately, since the primary author for this paper, Ms. Vrinda Gote is currently graduated and cannot perform experiments in the lab. An alternative to tube-formation assay, HUVEC cells migration assay is performed by the authors to assess the anti-angiogenic potential of OCT-Lcn2-Lipo.

  1. Why author have used condition medium from the MCF-7 and MDMBA231. Why they did not treat HUVEC cells with OCT-Lcn2-Lipo and Lcn2-Lipo for migration assay.

This is an excellent question raised by the reviewer. Conditioned media was used by the authors in this in vitro experiment for simulating internal environment, since the OCT-Lcn2-Lipo might not directly interact with the endothelial cells in the tumor microenvironment. But OCT-Lcn2-Lipo can interact indirectly by reducing the VEGF levels in the tumor surroundings. In addition to this, Lcn2 is well known to reduce angiogenesis, and hence reduce the growth and migration of HUVEC cells. Here we wanted to analyze the indirect effects of OCT-Lcn2-Lipo after it interacts with breast cancer cells, on angiogenesis of HUVEC cells. This is the reason for using conditioned media and using dual chamber in vitro HUVEC cells migration assay.

  1. Did the author look at the VEGF-C levels in MCF-7 as MCF-7 has high levels of VEGF-C.

This is an excellent suggestion by the reviewers. Unfortunately, since the primary author for this paper, Ms. Vrinda Gote is currently graduated and cannot perform experiments in the lab. In addition, I am retired and hence cannot perform this experiment. In this research article we primarily decided to evaluate the levels of VEGF-A, since it’s the most overexpressed isoform of VEGF. In addition, it is expressed in larger proportions as compared to other isoforms of VEGF.

  1. Authors have to improve on introduction and discussion and may be these articles will help 2020 Jun 21;9(6):1511; Cancer Epidemiol Biomarkers Prev. 2009 Feb;18(2):630-9. 

The authors are hoping that the reviewer can be more specific in what sections should be improved in the introduction and discussions.

The introduction discusses about breast cancer and its types, how metastasis occurs in breast cancer, current therapies and specifically siRNA therapies, difficulties to deliver siRNA, brief discussion about lipocalin-2 (Lcn-2), targeting agent octreotide and what we aim to achieve in this research.

While the discussion section discusses how targeting Lcn2 mRNA can reduce angiogenesis and metastatic in breast cancer, importance of each component of the liposome, detailed discussion of results, importance of formulation development and physiochemical analysis of the liposome and how the in vitro results relate with the primary aim of the research. This section also discusses examples of studies where scientists have utilized active tumor targeting with octreotide and utilized Lcn2 as an anti-angiogenesis agent.

Round 2

Reviewer 2 Report

Acepted